# A Multiagent Game Theoretic Simulation of Public Policy Coordination through Collaboration

Eleonora Herrera-Medina [1],* and Antoni Riera Font [2]

1 Centre for Economic Studies and Analysis, Antonio Nariño University, Circunvalar Campus, Bogotá 110237, Colombia
2 Doctoral School, University of the Balearic Islands, 07122 Palma, Spain; antoni.riera@uib.es
* Correspondence: director.ceae@uan.edu.co

**Abstract:** Background: Policy coordination is necessary to address many of the sustainability challenges we face today. The formal representations of policy coordination focus on modeling conflict management but neglect its collaborative nature. This limits efforts to build more realistic models of policy coordination. The objective of this paper is to simulate collaboration and noncollaboration between agents in the context of policy coordination in order to determine the effect of different approaches to policy coordination. Methods: For this purpose, a multiagent simulation of collaboration based on evolutionary game theory is used. Results: The results suggest that policy coordination through collaboration produces the most desirable outcomes and that reducing the cost of communication between agents is necessary to increase the probability of collaboration. Conclusions: The cost of information (both its transmission and transformation) is critical to increase the probability of collaboration in policy coordination. This paper advances the understanding of how to model the collaborative nature of policy coordination by contributing to the methodological standardization of the analysis and implementation of public policy coordination.

**Keywords:** collaboration; policy coordination; evolutionary game theory; multiagent systems; information cost

## 1. Introduction

Few sustainability challenges can be solved by the independent actions of individual policymakers; rather, their solutions require policy coordination between institutions. Through the coordination of public policies, the specialized contributions of different agencies and departments can be integrated [1]. Coordination enables the necessary dialogue and consensus between them, which allows them to agree to act according to certain rules or goals [2]. Without coordination, there may be a waste of resources, so coordination facilitates the exchange of resources, personnel, and knowledge between agencies [3]. Peters [4] also indicates that the coordination of public policies makes it possible to (i) avoid or minimize duplication and overlapping of policies, (ii) reduce policy inconsistencies, (iii) ensure policy priorities and aim for cohesion and coherence between them, (iv) mitigate political and bureaucratic conflict, and (v) promote a holistic perspective that goes beyond the narrow sectoral view of policies.

Even so, as Repetto [5] explains, coordination is not always an interactive process in which everyone who is involved wins but a process of seeking new equilibria where the results can be "zero sum": what the agent who leads the coordination wins is usually lost by those who must transfer the goods and/or services to be coordinated and that were previously under their sectoral responsibilities. Coordination is therefore a complicated phenomenon that involves strategic behavior and requires careful reasoning.

The preferred method of reasoning in economics is modeling [6]. Models enable policymakers to explain past coordination failures and successes and to predict the effects of current and proposed coordination efforts and from that derive recommendations. Due

to the complexity of coordination, there is a range of models that attempt to understand it. However, the problem with most current models of policy coordination is that the vast majority of them are qualitative [7–9] and present only general patterns supported by narrative explanations.

Qualitative models yield imprecise predictions about the directions of change and are often ambiguous, especially when multiple and conflicting cause-and-effect interactions are involved. They are an initial step in reasoning about a phenomenon, but further inquiry can only lead to reliable results if such models evolve into mathematical and quantitative models [10]. Through mathematical modeling, the consistency and precision of the reasoning is improved and, so, makes model evaluation more rigorous and lays a sounder basis for quantitative models [11]. Since the available models of policy coordination are not mathematical, their explanatory and predictive powers are limited.

This paper aims to develop a mathematical model of policy coordination with the aid of game theory, which is the appropriate framework for analyzing strategic behavior. This model is then converted into a simulation by means of multiagent modeling in order to analyze different approaches to policy coordination.

## 2. Literature Review

### 2.1. Theoretical Framework

In general, public policy coordination involves attempts to avoid conflicts between the decisions of different government agencies, as well as aligning such decisions and actions to produce solutions that are of mutual benefit to all [12]. Coordination can therefore be approached from the perspective of cooperation as a way to manage conflict or from the perspective of collaboration that is defined as a type of decision-making in which agents adjust their strategies for mutual benefit [13].

There have been attempts at the mathematical modeling of policy coordination mostly using game theory [14–16]. The game theoretic literature on the topic frames policy coordination from the perspective of coordination failure as caused by conflict. Game theory models on this topic therefore focus on understanding and avoiding conflict in order to reduce failures. Since game theory focuses on modeling rational approaches to conflict [17,18], it is well suited to this task.

Researchers prefer to employ noncooperative games to model policy coordination, and when they do so, we find that they fall into two categories. First, there are those associated with imbalances of incentives for cooperation (prisoner's dilemma) [9,19] and, second, those in which personal interest overrides common interest when there is social conflict over the use of limited resources (tragedy of the commons) [2,20]. Within the literature that employs noncooperative games, the models focus on situations in which agents have some conflicts of interest. By cooperating, they may choose an action that is not optimal for them but superior for society [21]. The emerging conflict between self-interest and social welfare leads to a social dilemma [22,23], and social dilemmas are at the root of many of the complex problems in public policy coordination, such as the efficient use of limited and scarce resources [20,24].

While all these models are useful to understand coordination strategies in general, they are not completely appropriate to understand public policy coordination. This is because the coordination of public policy in practice is largely the result of collaboration between agencies of the state, where conflict is not necessarily the predominant element [9]. Collaboration between decision-makers, based on consensus and collective trust, offers a way out of the problems generated by the prisoner's dilemma and other conflicts inherent in coordination [2]. This suggests that, instead of approaching coordination from the perspective of cooperation in the face of conflict, a more fruitful approach may be to look at it from the perspective of collaboration. This shifts the emphasis away from trying to reduce coordination failures toward increasing the likelihood of coordination successes.

Given its association with conflict, game theory has not sufficiently explored the modeling of collaboration, but more recent developments in game theory (especially those

found in population games and evolutionary games) offer useful tools to do this [25,26]. A first nonformal approximation of collaboration in evolutionary terms is found in [27], while [28] attempted to model collaboration with regard to the adoption of technology. However, it was Newton [29] who offered the first formal model of collaboration. While he demonstrated that game theory is a useful way to model collaboration, it is based on a primitive society without sophisticated agents or hierarchical institutional structures. He models coordination in a population of individual agents, whereas to model policy coordination we need a society that is composed of $m$ populations that, in turn, consist of groups and individuals. A more complex society requires individuals who are more sophisticated than in Newton's model, since they must be able to design public policies and coordinate them. Newton's model provides the theoretical basis for simulations of collaboration and related actions that allow experiments and the drawing of more specific behavioral conclusions and recommendations. With the appropriate modifications to this model, it will be possible to simulate a more complex and realistic society. By making these modifications, this paper is the first to offer a game theoretic model of policy coordination by means of collaboration.

It offers a unique perspective by recognizing the limitations of human cognitive abilities and behavioral biases in information processing [30]. By incorporating these costs into the model, the paper addresses the need to consider information processing costs in order to improve decision-making processes [31].

Furthermore, the paper emphasizes the role of collaboration in policy coordination. It argues that successful coordination requires collaboration between state agencies and suggests that models should focus on increasing coordination successes through collaboration rather than solely managing conflict and reducing coordination failures [9]. This perspective aligns with the modern approaches to governance that emphasize the importance of collaborative governance and multi-stakeholder engagement in addressing complex societal challenges [1,32].

### 2.2. Simulation Methods

Simulations offer a way to understand the impact of policy options, but given that the theoretical models of policy coordination emphasize conflict management, this understanding will be limited. Extending game theoretic models of policy coordination to include collaboration will make it possible to consider a wider range of options in simulations.

A promising new approach to simulate interactions between agents and the resulting complexity of economic systems is the multiagent simulation [33–35]. According to [36], multiagent systems are preferable to simple-agent-based models because they offer "a more emergent view of macroeconomic quantities". Multiagent modeling has already been used to simulate human and human-like behavior successfully in the fields of health care, education, decision systems [12,35,37], and engineering [15,38,39]. However, there has been no attempt in the economic literature to simulate policy coordination through collaboration using multiagent systems, so the multiagent simulation in this paper offers the first step in understanding the different approaches to policy coordination.

In several papers [33,34,40,41], evolutionary games were used as the theoretical basis for simulations of collaboration. This suggests that Newton's model [29] is a valid starting point from which to specify a simulation.

### 2.3. Contribution

In summary, the game theoretic models of policy coordination focus on managing conflict, whereas this paper extends the analysis to collaboration, which is neglected in the literature. While the literature of policy coordination recognizes that collaboration is critical to policy coordination, there are no studies that investigate the relative merits of collaborative and noncollaborative approaches or models of how such collaboration occurs.

The objective of this paper is to simulate collaboration and noncollaboration between agents in the context of policy coordination in order to determine the effect of different

approaches to policy coordination. This is undertaken by first identifying the adaptations to be made to Newton's game theoretic model of collaboration [29] and, second, by incorporating them into a multiagent simulation. The adaptations are then incorporated into an existing multiagent simulation that has already established the feasibility of collaboration in a noneconomic context [34].

We introduce a few minor modifications to enhance the realism of our simulation (in the following section), and the multiagent simulation is specified in the section thereafter. It employs three kinds of agents in three scenarios (noncoordination, coordination through cooperation, and coordination through collaboration). By comparing the outputs of these scenarios, the relative desirability of the different approaches to policy coordination will be inferred.

Multiagent modeling is a novel technique to research collaboration and its implications for establishing coordination in public policy. Another novel aspect of this research is that it includes information transmission and transformation costs in the model. These costs reflect humans' cognitive limits and behavioral biases in information processing, which have an impact on the success of collaboration and coordination efforts [6,8]. This modification improves the model's applicability to real-world policy implementation scenarios and provides a more accurate understanding of the problems encountered in attaining coordination [32,33]. Moreover, the research offers insights into the behavior of different types of agents and their interactions in a multiagent system. By analyzing the probabilities of collaboration and the impact of transmission and transformation costs, the study sheds light on the factors that influence successful coordination and the emergence of collaborative coordination [34,42].

This paper offers a more comprehensive understanding of collaboration and its implications for public policy coordination. It provides valuable insights into the dynamics of collaboration and the strategies that can enhance coordination outcomes. By identifying the factors that hinder or facilitate successful collaboration, the research contributes to the development of effective coordination strategies and the improvement of policy outcomes.

## 3. Materials and Methods

### 3.1. The Model to Be Simulated

In Newton's [29] model, there are no policies because the agents in his model do not think about the future. They are prehistoric humans whose only concern is hunting and seeking a safe haven for their clan, so they only think about the present. By introducing a collective problem that requires collective action, it becomes possible in this paper to make these agents more future oriented. This model will not simulate a modern policy environment but, rather, create a society in which policies can emerge as a result of collaboration. To introduce the possibility of policies, we need to introduce a third group of agents. These individuals are more sophisticated than Newton's, which means that they have the ability to share intentions, exhibit mutualistic behavior, think strategically, choose mixed strategies, and solve problems.

Given that the society in this model faces collective problems that need to be solved by collective action, there needs to be collaboration in order to find a solution by implementing policy-like solutions. So, unlike Newton's model, we assume that there is always the opportunity to collaborate, which makes the formulation of public policies more efficient. This ensures that there will always be collective problems to solve.

In [29], the balance condition implies that collaborative-type individuals will find themselves in groups in which collaboration occurs much more frequently. We must bear in mind that Newton deals with individuals who do not contemplate communities. The process of seeking solutions to collective problems occurs in a collaborative environment. Since collaboration is a mutualistic act, not an altruistic act, our group of problem solvers adjust their strategies and improve their payoffs, as well as the payoffs of their communities (solving common problems). Additionally, it is guaranteed that those who, for some reason, cannot solve problems do not adjust their strategies against themselves or their

communities. The balance condition guarantees the existence of at least 50% of individuals capable of solving problems in each community.

Since Newton's generality presents stability, the particular case is also in an evolutionarily stable state. So, in this society, groups of problem solvers evolve into institution-like organizations dedicated to improving the quality of life of their communities by solving collective problems and implementing policy-like solutions.

### 3.2. *The Game and the Multiagent System*

Gou and Deng [16] explored the process of evolutionary decision and stable strategies within multi-agent systems, including different types of agents involved in mutual collaboration based on evolutionary game theory. Their model was designed for the analysis of the consistency problem in robots and artificial intelligence, so it is necessary to make certain adaptations to use it in simulations of human societies.

#### 3.2.1. Adaptations

To simulate collaboration in human societies, a number of adaptations were made to Gou and Deng's model:

1.  Cost of transmitting and transforming information

To make the model of [19] more applicable to policies implemented by humans, information processing costs were added (which consist of the transmission cost and transformation cost of information). Information processing costs derive from humans' limited ability to process information, as several studies have shown [42–44].

Information processing goes through distinct phases: first information is transmitted by the senders, and then, it is transformed by the receivers. Since we are dealing with humans, and they have cognitive limitations or behavioral biases, each of these phases is susceptible to error [30,45,46]. This type of error is one of the causes of information processing costs.

During the processing of the information, we can incur information transformation costs, that is, make errors such as analysis errors or incorrect interpretations. In the same way, errors can be made during the information transmission process, which also leads us to incur costs. If the message/information transmitted is not correct or is incorrectly transmitted, the processed information will not fulfil its objective.

2.  Behavior of agents

The second adaptation that was made to the model has to do with the behavior of the cooperators. Unlike Gou and Deng's model, in this model, cooperators do not have the power to sabotage the system but only to react to incentives.

In our multiagent system, each agent can work individually or in a collaborative environment and interact with other agents. Agents that are characterized by mutualistic behavior deal with the formulation and solution of collective problems in a coordinated approach so that policies can emerge to achieve a common goal. The system can be affected when the information is transformed and transmitted, resulting in the possible failure of policy coordination to solve collective problems. We assume that these factors are described by the behavior of the collaborators, cooperators, and formulators.

#### 3.2.2. Interaction of Agents

Initially, the formulators transmit a message, a perceived problem, to both collaborators and cooperators. The collaborators analyze and interpret the information and, then, decide whether to share this transformed information with formulators while also messaging cooperators to invite them to collaborate. Cooperators decide whether or not to accept messages to collaborate.

All agents make free choices in the collaborative environment. The set of strategies for collaborators is to transform information or to not transform information. Whether they deliver whole or partial messages to the formulators, collaborators get messages and

earn rewards. The collaborators then determine whether or not the processed information is sent back to the formulators. They may get incentives for passing information back to the formulators.

For formulators, the strategy set is to send all messages or to send some messages. The formulators obtain benefits if their messages to collaborate are accepted, and these benefits are additional to the payoffs and do not imply costs. If their messages are not accepted, they incur a transmission cost.

The set strategy for cooperators is to accept the messages to collaborate or to not accept the messages to collaborate. For cooperators, if one of the messages sent by the formulators is incomplete, they receive a payoff, and if they decide to interact with formulators and collaborators, they receive a reward.

All these parameters are summarized in Table 1.

**Table 1.** Parameters of the game.

| Parameters | Description |
|---|---|
| Collaborators | |
| $P_{w1}, C_{w1} > 0$ | payments and costs associated with receiving messages from formulators |
| $P_{w2}, C_{w2} > 0$ | payments and costs associated with sending messages to cooperators |
| $B_w > 0$ | benefit for sharing the transformed information with formulators |
| $T_w > 0$ | transformation cost |
| $\alpha > 0$ | probability of sending transformed information to formulators |
| $\beta > 0$ | probability of receiving transmitted information from formulators |
| Formulators | |
| $P_{f1}, C_{f1} > 0$ | payments and costs associated with sending all messages to collaborate |
| $P_{f2}, C_{f2} > 0$ | payments and costs associated with sending only some messages |
| $Bf > 0$ | benefits for receiving a response to the message to collaborate |
| $Tf \geq 0$ | transmission cost |
| $0 > \rho > 0$ | probability of successful transmission of messages to collaborate |
| $\lambda$ | send all the messages |
| Cooperators | |
| $P_{B1}, C_{B1} > 0$ | payments and costs associated with receiving messages to collaborate from formulators |
| $P_{B2}, C_{B2} > 0$ | payments and costs associated with receiving messages to collaborate from collaborators |
| $U \geq 0$ | payments for receiving failed messages |
| $0 > \gamma > 0$ | probability of collaboration when they successfully receive a message |
| $R_w, R_f > 0$ | reward for collaborating with formulators and collaborators |

### 3.2.3. The Payoff of Agents

In [29], we assume that half of our population are collaborators ($x = 0.5$), fewer are formulators ($y = 0.3$), and cooperators are the minority with ($z = 0.2$). For collaborators, $x(0 \leq x \leq 1)$ is the probability that the collaborators accept the messages, which implies that the probability of nonacceptance is $(1 - x)$ For formulators, $y(y = 1)$ represents the number of messages, while $0 < y < 1$ indicates the number of incompletely sent messages. For cooperators, the probability of receiving messages is $z$, and $(1 - z)$ denotes the probability of messages they do not receive.

The expected payoffs are obtained from the payoff matrix in Table 2 and the strategies chosen by each of the agents (see Appendix A for equations).

**Table 2.** Payoff matrix for Formulators, Collaborators, and Cooperators.

| | All (y = 1) | | Some (0 < y < 1) | |
|---|---|---|---|---|
| | Accept (z) | Not Accept (1 − z) | Accept (z) | Not Accept (1 − z) |
| Transform (x) | $(W_1, F_1, B_1)$ | $(W_2, F_2, B_2)$ | $(W_3, F_3, B_3)$ | $(W_4, F_4, B_4)$ |
| Not transform (1 − x) | $(W_5, F_5, B_5)$ | $(W_6, F_6, B_6)$ | $(W_7, F_7, B_7)$ | $(W_8, F_8, B_8)$ |

3.2.4. Expected Payments and the Replicator Dynamics

Equation (1) shows the expected average payoffs for the collaborators, and it is composed of the expected payoffs of the collaborators when they send the transformed information plus the expected payoffs of not sending it:

$$\overline{E_x} = xE_x + (1 - x)E_{1-x} \tag{1}$$

From Equation (1), we can obtain Equation (2), which is the replicator dynamic equation of a collaborator:

$$F_x = x(1 - x)[\lambda + y(1 - \lambda)](\alpha B_w - T_w) \tag{2}$$

Equation (3) shows the expected average payoffs for the formulators, and it is composed of the expected payoffs of sending all the messages plus the expected payoffs of sending only some of the messages:

$$\overline{E_y} = yE_y + (1 - y)E_{1-y} \tag{3}$$

From Equation (3), we can obtain Equation (4), which is the replicator dynamic equation of the formulators:

$$F_y = y(1 - y)\left[\rho\left(P_{f1} - P_{f2}\right) + \left(C_{f2} - C_{f1}\right) + x(\beta Bf - Tf)(1 - \lambda)\right] \tag{4}$$

Equation (5) shows the average expected payoffs for the cooperators, and it is composed of the expected payoffs when they receive the messages plus when they do not receive the messages:

$$\overline{E_z} = zE_z + (1 - z)E_{1-z} \tag{5}$$

Similarly, from Equation (5), we can obtain Equation (6), which is the replicator dynamic equation for the cooperators:

$$F_z = z(1 - z)\left\{[\beta(P_{B1} + P_{B2}) - (C_{B2} + C_{B1})]\left[\lambda + y\left((1 + \lambda) + \gamma\left(R_w + R_f\right) - U\right)\right]\right\} \tag{6}$$

These replicator equations, together with the strategies, show us the dynamic process of convergence toward the steady state. From these results, we run the simulation for the different parameters in order to analyze their influence on the probability of collaboration under the constraint conditions of evolutionary stable strategies (ESS) and, from there, generate our three scenarios.

## 4. Results

*4.1. The Simulation and the Three Scenarios*

Three scenarios will be simulated: noncoordination, coordination through cooperation, and coordination through collaboration. The relative desirability of the various approaches to policy coordination will be inferred by comparing the outputs of these scenarios. These three scenarios are the evolutionary result of the equilibrium points under the constraint conditions of the ESS.

### 4.1.1. Scenario 1—Noncoordination

A noncoordination scenario is one in which the agents do not interact. There is no exchange of information between agents. As there is no exchange of information, it is impossible to initiate collaboration and/or cooperation; therefore, coordination is not possible.

For this simulation, we assign initial values to the different parameters (See Table 1) based on the stability and constraint conditions (Equations (7)–(9)):

$$\alpha B_w - T_w < 0 \tag{7}$$

$$\rho\left(P_{f1} - P_{f2}\right) + \left(C_{f2} - C_{f1}\right) < 0 \tag{8}$$

$$\lambda[\beta(P_{B1} + P_{B2}) + (C_{B1} - C_{B2})] < U - \gamma\left(R_w + R_f\right) \tag{9}$$

Figure 1 shows the results of the simulation for the noncoordination scenario. The probability of collaboration appears on the x-axis and the number of rounds on the y-axis.

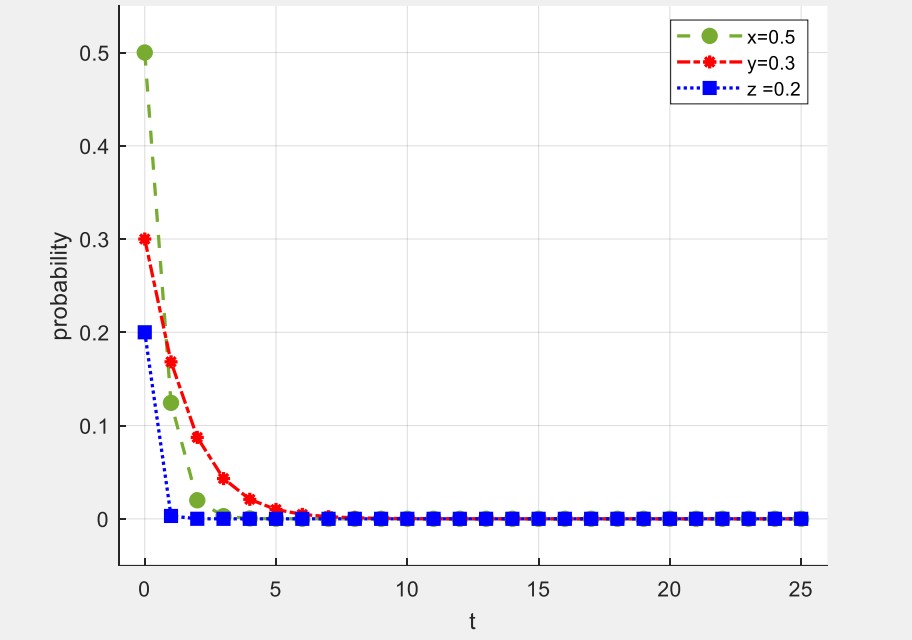

**Figure 1.** Scenario 1—noncoordination.

The first thing to notice is that all strategies tend to zero; in fact, the system converges rapidly to zero. After five rounds, the strategies tend to zero, which means that there is no collaboration. The foregoing implies that, if it is very costly for the formulators to send the initial messages, neither the collaborators nor the cooperators can interact with each other, and as a consequence, there is no possibility of coordination.

### 4.1.2. Scenario 2—Coordination through Cooperation

A scenario of coordination through cooperation is one in which the exchange of information exists but not among all the agents. One of the agents exchanges information only when it receives incentives, and one of the agents loses interest and/or decides to stop exchanging information in response to the behavior of the agent that responds only to incentives.

For the simulation, we assign initial values to the different parameters on the stability and constraint conditions (Equations (10)–(12)):

$$\alpha B_w - T_w > 0 \tag{10}$$

$$\rho\left(P_{f1} - P_{f2}\right) + \left(C_{f2} - C_{f1}\right) < \left[\beta B_f - T_f\right](\lambda - 1) \tag{11}$$

$$\lambda[\beta(P_{B1} + P_{B2}) + (C_{B1} - C_{B2})] > U - \gamma\left(R_w + R_f\right) \tag{12}$$

Figure 2 shows the results of the simulation for coordination through cooperation.

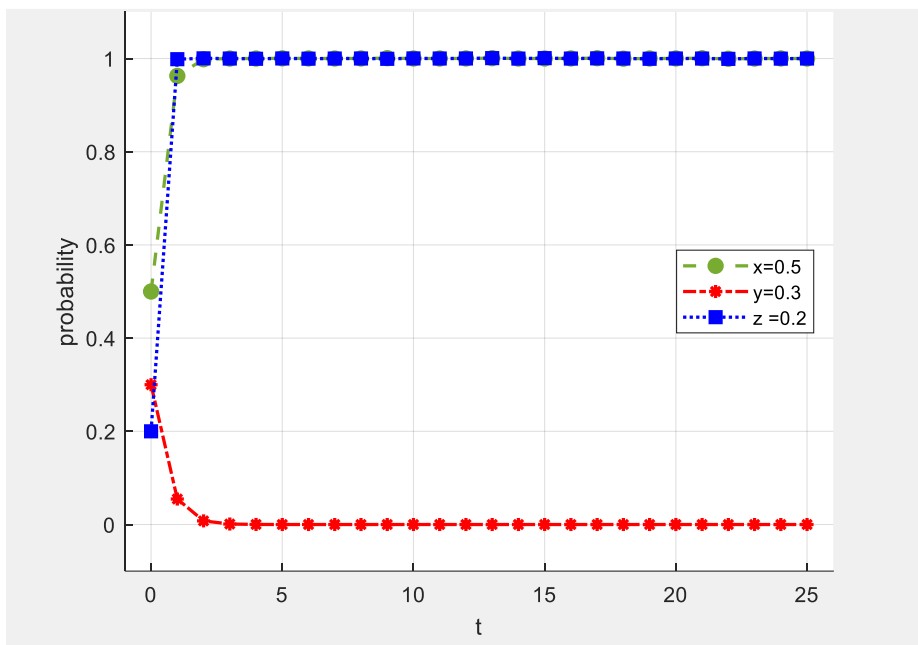

**Figure 2.** Scenario 2—coordination through cooperation.

The strategies of the collaborators and cooperators converge quickly to one, while the strategies of the formulators converge to zero but slowly. This indicates an active interaction between these two agents, as well as an active exchange of information. This behavior allows for the possibility of coordination through cooperation between agents, but it does not involve all the agents that are part of the system. We can observe that the formulators lose interest in the interaction and do not participate in the exchange of information.

4.1.3. Scenario 3—Coordination through Collaboration

A scenario of coordination through collaboration is one in which information flows between all agents. The strategies of the agents are aligned toward the achievement of a common objective.

For the simulation, we assign initial values to the different parameters on the stability and constraint conditions (Equations (13)–(15)):

$$\alpha B_w - T_w > 0 \tag{13}$$

$$\rho\left(P_{f1} - P_{f2}\right) + \left(C_{f2} - C_{f1}\right) > \left[\beta B_f - T_f\right](\lambda - 1) \tag{14}$$

$$\beta(P_{B1} + P_{B2}) + (C_{B1} - C_{B2}) > U - \gamma\left(R_w + R_f\right) \tag{15}$$

Figure 3 shows the results of the simulation for the coordination through collaboration scenario.

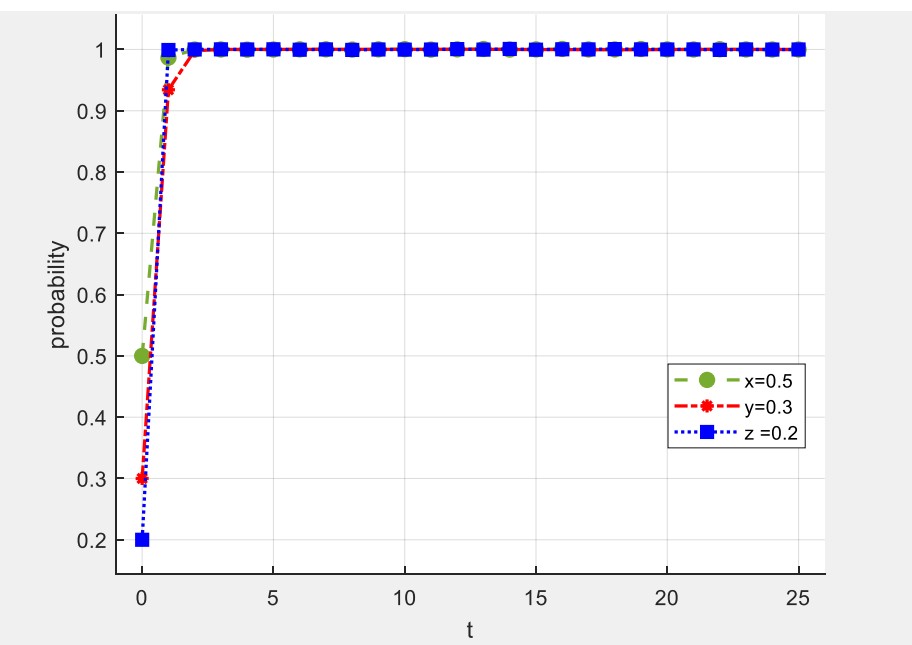

**Figure 3.** Scenario 3—coordination through collaboration.

As we can see, all the strategies quickly converge to one, which indicates an exchange and active processing of information and interaction between agents, giving rise to coordination through collaboration. This scenario is theoretically and empirically ideal, but it is rarely achieved.

### 4.2. Achieving Collaboration

The three previous scenarios showed us the behavior of all the parameters at the same time. Let us analyze how these transformation costs ($T_F$) and transmission costs ($T_W$) of information affect the parameters ($\alpha$, $\beta$, $\rho$, $\gamma$) and the system, bearing in mind that information processing costs can negatively affect the ability of agents to process information.

#### 4.2.1. $\alpha$ Probability of Sending Information

This parameter measures the probability that the three groups of agents will send transformed information.

In Figure 4a, we can observe that $\alpha$ affects the strategy selection of collaborators. As the probability of sending transformed information ($\alpha$) increases, the chance of collaboration increases. Transformation costs ($T_F$) affect the probability of sending information ($\alpha$). In Figure 4b, we can observe how by eliminating the transformation costs, the increase in the probability of sending the transformed information and the effect of $\alpha$ on the choice of strategies of the collaborators bring the system to the point of coordination through collaboration.

In Figure 4c, we can observe that $\alpha$ affects the strategy selection of formulators. It also shows that the increase in the probability of sending the transformed information improves the possibilities of the collaboration of formulators. In Figure 4d, we can observe that zero transmission costs speed up the coordination process through collaboration.

Collaborators

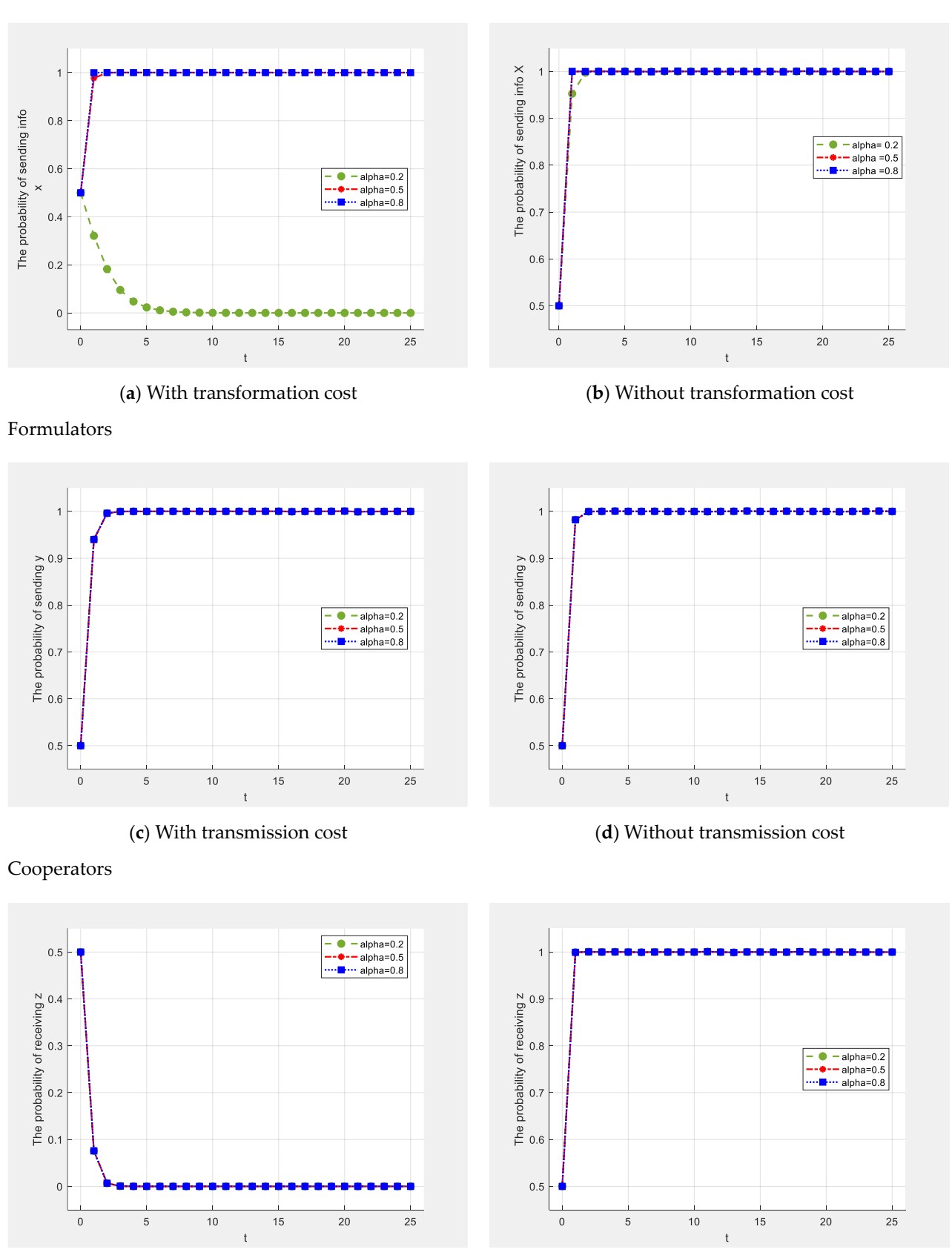

(**a**) With transformation cost

(**b**) Without transformation cost

Formulators

(**c**) With transmission cost

(**d**) Without transmission cost

Cooperators

(**e**) With transmission cost

(**f**) Without transmission cost

**Figure 4.** $\alpha$ Probability of Sending Information.

In Figure 4e, we can observe that $\alpha$ has no effect on the strategy choice of cooperators. Figure 4f shows that when transmission costs are equal to zero, this implies that there are no problems during the transmission of information, so there are no errors during transmission. That is, when Tw = 0, U = 0, which makes the system for the cooperators to be of the coordination-through-collaboration type.

### 4.2.2. $\beta$ Probability of Receiving Information

This parameter allows us to measure the probability that agents receive transmitted information.

In Figure 5a, we can observe that $\beta$ does not affect the selection of strategies by the collaborators when the transformation costs are greater than zero. In Figure 5b, we can observe how transformation costs equal to zero make the system for the collaborators to be of the coordination-through-collaboration type. Transformation costs affect the behavior of $\beta$.

Collaborators

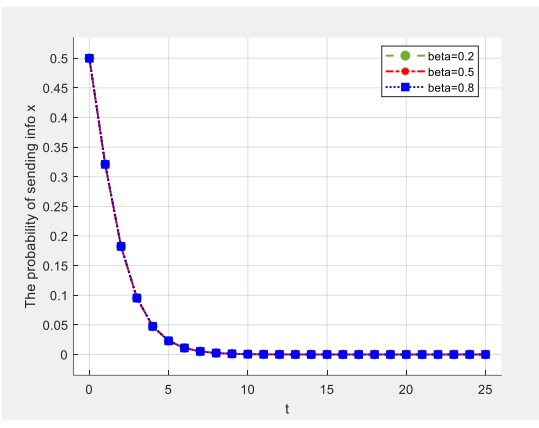

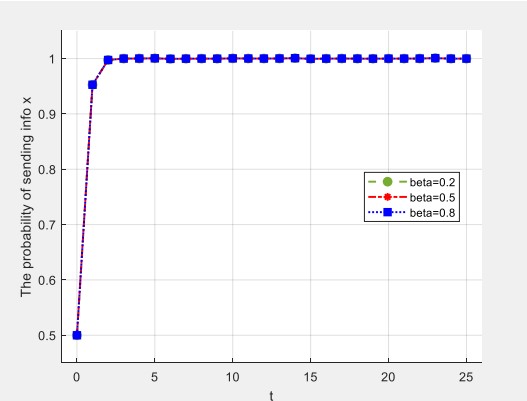

(**a**) With transformation cost          (**b**) Without transformation cost

Formulators

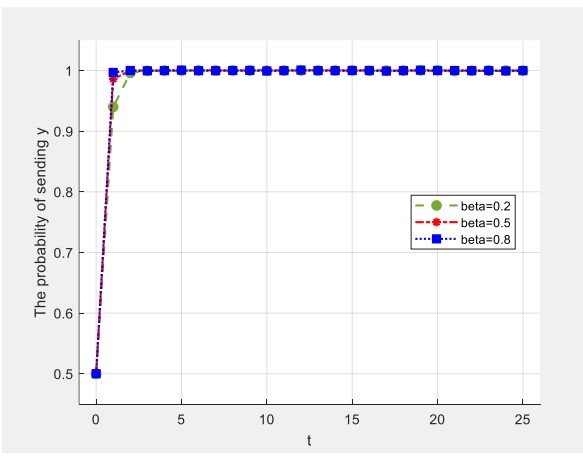

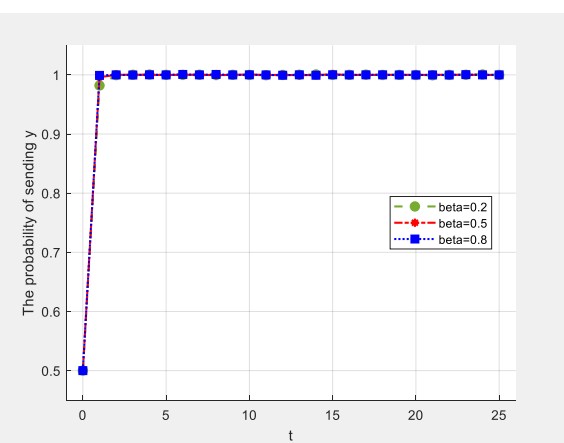

(**c**) With transmission cost          (**d**) Without transmission cost

**Figure 5.** *Cont.*

Cooperators

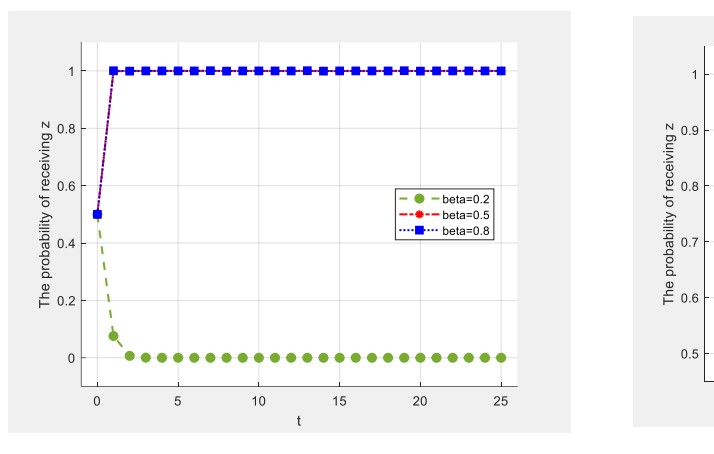
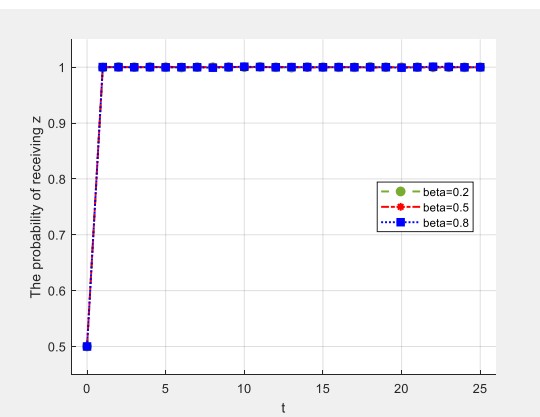

(**e**) With transmission cost

(**f**) Without transmission cost

**Figure 5.** β Probability of Receiving Information.

In Figure 5c, we can observe that β affects the strategy selection of formulators. It also shows that the increase in the probability of receiving the transformed information improves the possibilities of the collaboration of formulators. In Figure 5d, we can observe how zero transmission costs speed up the coordination process through collaboration.

In Figure 5e, we can observe that β affects the strategy selection of Cooperators. It is also shown that as the probabilities of receiving information increase, the agents' possibilities for collaboration improve. In Figure 5f, we can also observe that by eliminating the transmission costs, the increase in the probability of receiving information and the effect of β on the choice of the strategies of the cooperators, the system is brought to the ideal of coordination through collaboration.

### 4.2.3. ρ Probability of Successful Transmission

This parameter allows us to measure the probability of successful transmission of messages inviting collaboration.

In Figure 6a, we can observe that ρ affects the strategy selection of formulators. It is also shown that as the probabilities of successful transmission increase, the agents' possibilities for collaboration improve.

Formulators

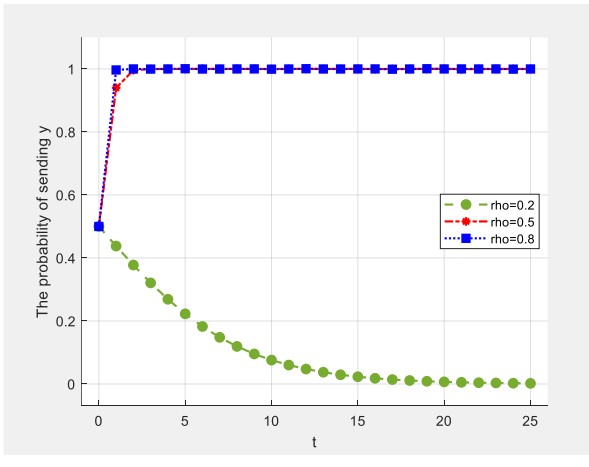
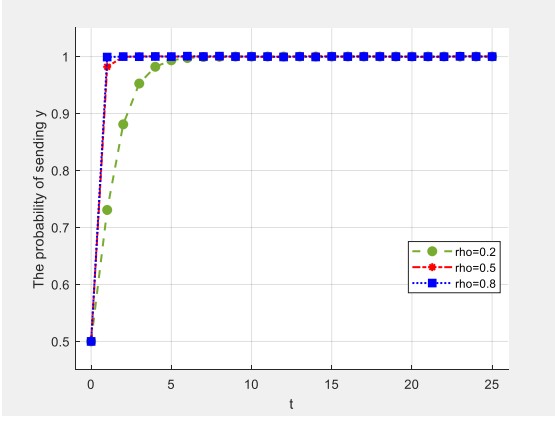

(**a**) With transformation cost

(**b**) Without transformation cost

**Figure 6.** *Cont.*

Collaborators

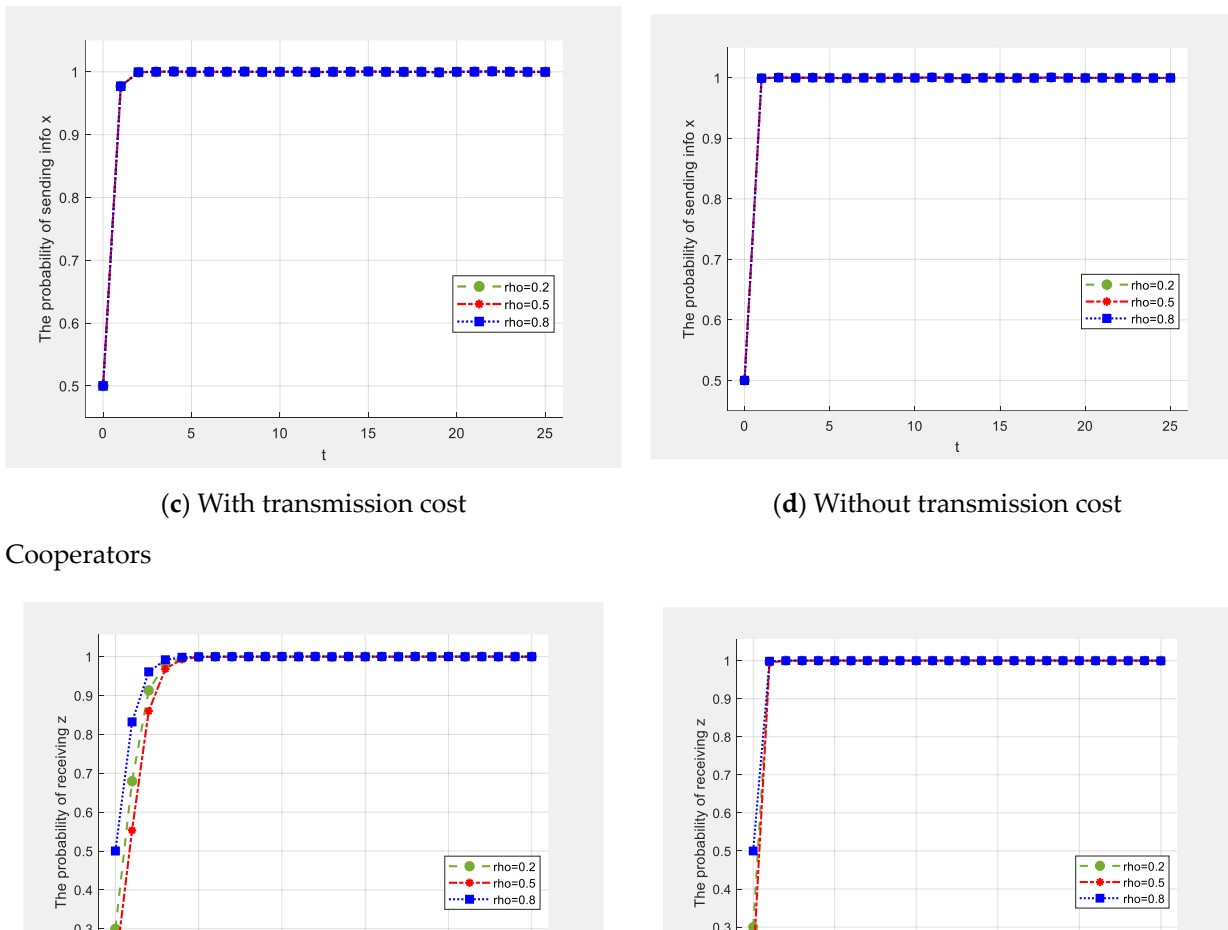

(**c**) With transmission cost

(**d**) Without transmission cost

Cooperators

(**e**) With transmission cost

(**f**) Without transmission cost

**Figure 6.** $\rho$ Probability of Successful Transmission.

In Figure 6b, we can also observe that by eliminating the transmission costs, the increase in the probability of successful transmission, and the effect of $\rho$ on the choice of strategies of the formulators, the system is brought to the ideal of coordination through collaboration.

In Figure 6c, we can observe that $\rho$ affects the strategy selection of collaborators. In Figure 6d, we can observe how zero transformation costs speed up the coordination process through collaboration.

In Figure 6e, we can observe that $\rho$ affects the strategy selection of cooperators and how transmission costs slow down the process of coordination through collaboration. In Figure 6f, we can observe how zero transmission costs speed up the coordination process through collaboration.

### 4.2.4. $\gamma$ Probability of Collaboration

This parameter allows us to measure the probability of collaboration once agents successfully receive the messages.

In Figure 7a, we can observe that $\gamma$ affects the strategy selection of collaborators and how transformation costs slow down the process of coordination through collaboration.

In Figure 7b, we can observe how zero transformation costs speed up the coordination process through collaboration.

Collaborators

Formulators

Cooperators

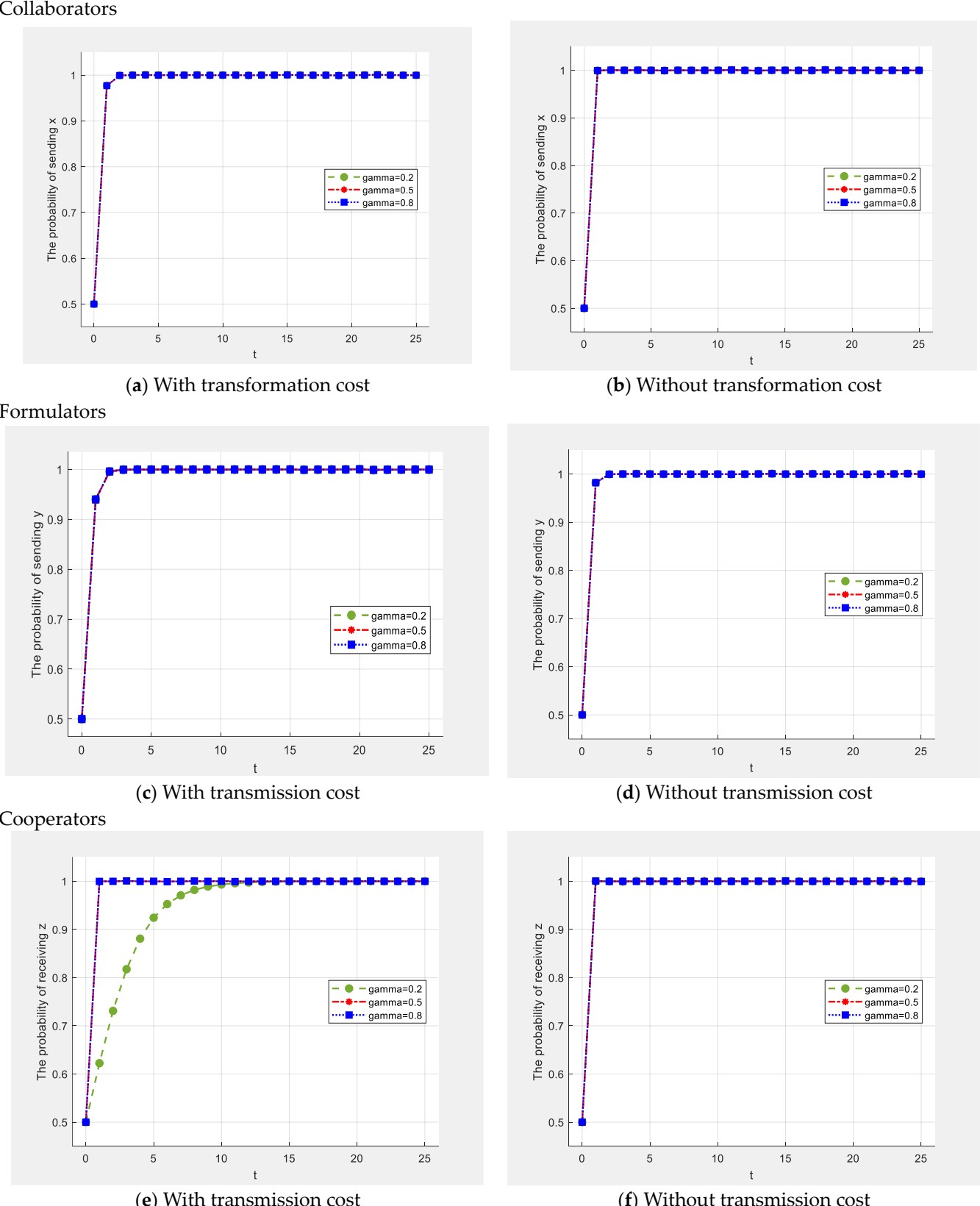

**Figure 7.** $\gamma$ Probability of Collaboration.

In Figure 7c, we can observe that γ affects the strategy selection of formulators and how transmission costs slow down the process of coordination through collaboration. In Figure 7d, we can observe how zero transmission costs speed up the coordination process through collaboration.

In Figure 7e, we can observe that γ affects the strategy selection of cooperators. It is also shown that, as the probabilities of collaboration once they successfully receive the messages increase, the agents' possibilities for collaboration improve.

We can also see that, by eliminating the transmission costs in Figure 7f, the increase in the probability of collaboration once they successfully receive the messages, and the effect of γ on the choice of strategies of the cooperators, the system is brought to the ideal of coordination through collaboration.

## 5. Discussion

The simulation results indicate that the formulators play a crucial role in collaboration. If the formulators successfully transmit all messages ($T_w = 0$), they offer a fundamental assurance for the sharing of reliable information. Inversely, failure to deliver full information results in a deviation from common goals and affects how collaborators respond ($T_f > 0$). In a multiagent system, collaboration is essential to ensuring that all agents create a cohesive whole in an anticipatory manner. In practice, evidence is neither neutral nor a quality that can be retrieved from a shelf. Producing and using evidence strategically is possible [47].

Additionally, according to the simulation outcomes, we determined that the probability of collaboration is ideal when the agents agree to follow a common information-sharing objective. That is, formulators transmit all communications, collaborators respond and send messages on time, and cooperators favorably accept messages, so the probabilities of sending (α) and receiving (β) information also increase the probabilities of collaboration. This result suggests that our proposed model can be applied in practice.

In terms of information cost, the results of the modeling provide insights into the impact of information costs on policy coordination. The results show the effects of transmission cost ($T_w$) and transformation cost ($T_f$) on the probability of sending and receiving information, as well as the strategies adopted by different agents.

The findings indicate that information costs play a significant role in shaping the behavior and outcomes of policy coordination. When transmission costs are high, the probability of sending and receiving information decreases, leading to a lower likelihood of collaboration. This suggests that reducing transmission costs, such as improving communication channels and information-sharing platforms, is crucial for enhancing policy coordination [42,44,45,48].

Similarly, transformation costs have an impact on the probability of sending information. Higher transformation costs reduce the probability of sending transformed information, which hinders collaboration among agents. This highlights the importance of minimizing transformation costs, such as by improving information-processing systems and reducing errors in information transformation to facilitate effective policy coordination [42,44,45,48].

Furthermore, the results demonstrate that reducing information costs can lead to more desirable coordination outcomes. When transmission costs are eliminated ($Tw = 0$), the system moves toward coordination through collaboration, where all agents actively exchange information and collaborate. Similarly, when transformation costs are eliminated ($T_f = 0$), the system shifts toward coordination through collaboration, indicating a higher probability of collaboration among agents.

These findings underscore the significance of addressing information costs in policy coordination efforts. By reducing transmission and transformation costs, policymakers can enhance the probability of collaboration, improve information exchange, and achieve more effective coordination outcomes [45,47,49]. This highlights the need for investments in information systems, technology, and training to streamline information processing and communication in policymaking and governance.

In summary, the results of the modeling highlight the influence of information costs on policy coordination. By reducing transmission and transformation costs, policymakers can enhance collaboration, improve information exchange, and achieve more effective coordination outcomes in the pursuit of sustainable policy objectives.

Through this model, one shortcoming in the formal treatment of policy coordination has been addressed, a shortcoming that [3] describes as follows: "there is still no standardized method for approaching coordination issues, and much of the success or failure of attempts to coordinate appears to depend upon context". The model presented in this paper contributes to the methodological standardization of the analysis and implementation of public policy coordination.

## 6. Conclusions

Almost all of today's sustainability concerns necessitate policy coordination; nevertheless, research on the modeling of such coordination is lacking in public policy research. The objective of this paper was therefore to simulate collaboration and noncollaboration between agents in the context of policy coordination in order to determine the effect of different approaches to policy coordination. For this purpose, a multiagent simulation of collaboration based on evolutionary game theory was used.

The paper proposes that collaboration across state agencies is a crucial driver of successful policy coordination and that it should focus on enhancing coordination successes through collaboration rather than managing conflict and decreasing coordination failures [18,20]. To handle complex societal concerns, modern governance emphasizes collaborative coordination and multi-stakeholder engagement.

The results suggest that policy coordination through collaboration produces the most desirable outcomes and that reducing the cost of communication between agents is necessary to increase the probability of collaboration. The research recognizes the limitations of human cognitive capacities and behavioral biases by integrating the cost of transmitting and transforming information into the model [6,7]. This awareness is critical in the modernization of policy-making processes because it underlines the importance of accounting for the obstacles and costs of information processing in order to improve decision-making and coordination. The cost of information (both its transmission and transformation) is critical to increase the probability of collaboration in policy coordination. This paper advances the understanding of how to model the collaborative nature of policy coordination by contributing to the methodological standardization of the analysis and implementation of public policy coordination.

Reduced information processing costs are required for policy implementation. Complex and interconnected issues necessitate the integration of disparate information sources and the coordination of operations across sectors and levels of government [35,38,46]. Policymakers can improve the efficiency and success of sustainability programs by tackling information transmission and transformation costs.

Overall, the findings of this article have implications for enhancing policy-making processes by accounting for the costs of information processing, prioritizing collaboration in policy coordination, and creating standardized approaches for analyzing and implementing public policy coordination.

Policymakers and government agencies can use the findings to create more effective coordination methods and improve the results of their policy endeavors [18]. Collaborators and cooperators can better grasp the aspects that drive successful collaboration and work together to achieve common goals [31,34]. Furthermore, researchers and academics can build on this work to further investigate the dynamics of collaborative coordination and their applications.

**Author Contributions:** E.H.-M. is the primary researcher in this project. A.R.F. provided supervision and assisted with the conceptualization, model validation, and review and editing of the research. All authors have read and agreed to the published version of the manuscript.

**Funding:** This research received no specific grant from any funding agency in the public, commercial, or not-for-profit sectors.

**Institutional Review Board Statement:** Not applicable.

**Informed Consent Statement:** Not applicable.

**Data Availability Statement:** Data sharing is not applicable to this article as no new data were created or analyzed in this study. The MATLAB code used for the simulations is available upon request.

**Conflicts of Interest:** The authors declare no conflict of interest.

**Appendix A**

Payoffs of different strategies:

$$W_1 = \beta Pw_1 + \beta Pw_2 - Cw_1 - Cw_2 + \alpha Bw - Tw$$

$$F_1 = \rho P_{f1} - Cf_1 + \beta Bf - Tf$$

$$B_1 = \beta P_{B1} + \beta P_{B2} - C_{B1} - C_{B2} + \gamma \left[ R_w + R_f \right]$$

$$W_2 = 0$$

$$F_2 = \rho P_{f_1} - C_{f1} + \beta Bf - Tf$$

$$B_2 = 0$$

$$W_3 = \lambda (\beta P_{w1} + \beta P_{w2} - C_{w1} - C_{w2} + \alpha B_w - T_w)$$

$$F_3 = \rho P_{f2} - C_{f2} + \lambda (\beta Bf - Tf)$$

$$B_3 = \lambda (\beta P_{B1} + \beta P_{B2} - C_{B1} - C_{B2}) + \gamma \left[ R_w + R_f \right]$$

$$W_4 = \lambda (\beta P_{w1} + \beta P_{w2} - C_{w1} - C_{w2} + \alpha B_w - T_w)$$

$$F_4 = \rho P_{f2} - C_{f2} + \lambda (\beta Bf - Tf)$$

$$B_4 = U$$

$$W_5 = \beta P_{w1} + \beta P_{w2} - C_{w1} - C_{w2}$$

$$F_5 = \rho P_{f1} - C_{f1}$$

$$B_5 = \beta P_{B1} + \beta P_{B2} - C_{B1} - C_{B2} + \gamma \left[ R_w + R_f \right]$$

$$W_6 = \beta P_{w1} + \beta P_{w2} - C_{w1} - C_{w2}$$

$$F_6 = \rho P_{f1} - C_{f1}$$

$$B_6 = U$$

$$W_7 = \lambda(\beta P_{w1} + \beta P_{w2} - C_{w1} - C_{w2})$$

$$F_7 = \rho P_{f2} - C_{f2}$$

$$B_7 = \lambda(\beta P_{B1} + \beta P_{B2} - C_{B1} - C_{B2}) + \gamma \left[ R_w + R_f \right]$$

$$W_8 = \lambda(\beta P_{w1} + \beta P_{w2} - C_{w1} - C_{w2})$$

$$F_8 = \rho P_{f2} - C_{f2}$$

$$B_8 = U$$

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
