# Peer review of "A Multiagent Game Theoretic Simulation of Public Policy Coordination through Collaboration"

_sustainability, doi:10.3390/su151511887_

Round 1
Reviewer 1 Report
This paper is interesting to read and contributed significantly to the body of public policy literature. The following suggestions should be carefully addressed by the author(s) in order to make the study more obvious in accordance with conventional research procedures and to raise the article's quality in accordance with the journal's research principles:
The authors have to justify the topic selection.
Increase the study's contribution and indicate the reference to its uniqueness.
Improve the literature review section by including recent research.
https://doi.org/10.1016/j.scib.2021.11.010
https://doi.org/10.1016/j.ijpe.2021.108166
https://doi.org/10.1007/s00521-020-04958-9
https://doi.org/10.1016/j.landusepol.2022.106379
https://doi.org/10.1016/j.econmod.2023.106194
Some of the policy proposals are general; the author should provide recommendations that are unique to public policy.
The papers include several grammatical problems and need further editing.
Reviewer 2 Report
This paper uses multi-agent simulation of collaboration to simulate collaboration and non-collaboration between agents in the context of policy coordination to determine the relative desirability of different approaches to policy coordination. However, the authors do not clearly disclose the significance or meaning of this research. There are lots of essential parts need to be clarified and some sections is suggested to rewrite since there too many repeated wording and sentences. 1. Eliminate multiple references. After that please check the manuscript thoroughly and eliminate all the lumps in the manuscript. This should be done by characterizing each reference individually. This can be done by mentioning 1 or 2 phrases per reference to show how it is different from the others and why it deserves mentioning. 2. In the introduction, you need to connect the state of the art to your paper goals. Please follow the literature review by a clear and concise state of the art analysis. This should clearly show the knowledge gaps identified and link them to your paper goals. Please reason both the novelty and the relevance of your paper goals. 3. The reader is left without detailed knowledge on what is known on the topic based on prior studies and what needs to be known. A focused discussion would also enable the authors to state the contribution of the paper more clearly. 4. The figures size is inconsistent. (Figure 4c-Figure 7f) 5. I read the results and discussion section completely. The discussion section is the main part of a paper, but this manuscript mainly reported the data of the modelling without discussing it through adding available reasoning for justifying the result. I recommend author adding several reasoning and comparison through available publications in the literature. 6. In the conclusions, in addition to summarizing the actions taken and results, please strengthen the explanation of their significance. lt is recommended to use quantitative reasoning comparing with appropriate benchmarks, especially those stemming from previous work. 7. Please insert a section on the implications of the study. Who benefits with it? What problem can the study help to solve? What’s next? 8. The conclusion part should be more refined to make the findings and contributions of the paper clearer. Furthermore, please note the difference between the conclusions and abstract. 9. So, a clearer illustration of contribution or innovation should be further provided in the introduction and conclusion. 10. Adding some relevant countermeasures and suggestions. 11. My first and primary concern lies in the novelty of this work, as I feel that the novelty issue has not been sufficiently highlighted in the current version. An important question shall be answered: does this work fill up some knowledge gaps which previous articles cannot address? 12. The discussion is related to the theory, but the relevance of the findings to the modernization of the state of art is not clear. The methodology is well designed, but there are missing elements that relate the proposed to what was found by the authors. Contributions are unclear.
Moderate editing of English language required.
Reviewer 3 Report
The piece describes a curious study that aimed to simulate collaboration and non-collaboration between agents in the context of policy coordination to determine the relative desirability of different approaches to policy coordination. The article is interesting; however, there are a few issues that hindered the manuscript, which initially appeared promising, resulting in some disappointment. I will attempt to clarify what puzzles me:
- Abstract: the background and purpose are understood, but the method should be described in more detail it is currently too ambiguous. Additionally, it is unclear what value this analysis provides for readers of this journal of educational research.
- Introduction: the literature review barely mentions relevant studies and is too short. As a result, a thorough discussion of the results cannot be effectively conducted afterwards. It seems that the authors themselves may have faced challenges in connecting the background to the results.
- Methodology: this section is the most disappointing. It is not well understood, and many questions arise. Has it been validated? Is it reliable? Without this information, the rigor of the study remains questionable.
- Discussion and conclusions: these sections are very brief. There is little engagement with previous studies, possibly due to a lack of a comprehensive theoretical framework, which is reflected in this section.
- Limitations, practical implications and possible future lines of research are not addressed.
- Results: there is an excessive of formulas and figures that make it difficult to comprehend the article.
- The discussion becomes vague. This could be attributed to the inadequate literature review, which creates a disconnect between the introduction and the results.
- Finally, the citations in the text do not follow the referencing style used by the journal. References are also missing.
Round 2
Reviewer 1 Report
no more comments.
Author Response
Thank you for your assistance in improving our paper.
Reviewer 3 Report
Good job. You have managed to address the issues raised and improved the introduction and conclusion, which were the two most disappointing sections of the article. However, I still see that a consistent referencing style, in accordance with the journal's guidelines, is not being followed. For example, references number 10, 33, and 35 (just to name a few) follow a different style compared to the rest. It would be good to unify it before the article is published.
Author Response
Thank you. I checked all the references again, and corrected them.